# Screening and Diversity Analysis of Aerobic Denitrifying Phosphate Accumulating Bacteria Cultivated from A$^2$O Activated Sludge

**Yong Li [1,\*], Siyuan Zhao [1], Jiejie Zhang [2,3,4], Yang He [1], Jianqiang Zhang [1] and Rong Ge [5,\*]**

[1] Faculty of Geosciences and Environmental Engineering, Southwest Jiaotong University, Chengdu 610059, China; 646761704@163.com (S.Z.); yanghe@swjtu.cn (Y.H.); zhjiqicn@swjtu.cn (J.Z.)

[2] Research Center for Eco-Environmental Sciences, Chinese Academy of Sciences, Beijing 100085, China; txgsfy@163.com

[3] Sino-Dansih College, University of Chinese Academy of Sciences, Beijing 101400, China

[4] Sino-Danish Center for Education and Research, University of Chinese Academy of Sciences, Beijing 100190, China

[5] Navigation College, Jiangsu Maritime Institute, Nanjing 211170, China

\* Correspondence: liyong@swjtu.edu.cn (Y.L.); grsmu@163.com (R.G.);
Tel.: +86-135-1810-8466 (Y.L.); +86-159-5199-6696 (R.G.)

**Abstract:** The aerobic denitrifying phosphate accumulating bacteria (ADPB) use $NO_3^-$ as an electron acceptor and remove nitrate by denitrification and concomitant uptake of excessive phosphorus in aerobic conditions. Activated sludge was collected from the A$^2$O aerobic biological pool of the sewage treatment plant at Hezuo Town, Chengdu City. The candidate ADPB strains were obtained by cultivation in the enriched denitrification media, followed by repeated isolation and purification on bromothymol blue (BTB) solid plates. The obtained candidates were further screened for ADPB strains by phosphorus uptake experiment, nitrate reduction test, metachromatic granules staining, and poly-β-hydroxybutyrate (PHB) staining. The 16 sedimentation ribosome deoxyribonucleic acid (16 S rDNA) molecular technique was used to determine their taxonomy. Further, the denitrification and dephosphorization capacities of ADPB strains were ascertained through their growth characteristics in nitrogen-phosphorus-rich liquid media. The results revealed a total of 25 ADPB strains screened from the activated sludge of the A$^2$O aerobic biological pool of the sewage treatment plant at Hezuo Town. These strains belonged to two classes, four orders, and five genera. Among them, the strain SW18NP2 was a potentially new species in the *Acinetobacter* genus, while the strain SW18NP24 was a potential new species in the *Pseudomonas* genus at the time of their characterization. The *Acinetobacter* was the dominant genus. The obtained ADPB strains demonstrated a rich diversity. The ADPB strains had significant variations in denitrification and dephosphorization capacities. Twenty-three strains exhibited a total phosphorus removal rate of above 50%, and 19 strains exhibited a total nitrogen removal rate of above 50%. The strain SW18NP2 showed the best denitrifying phosphorus removal (DPR) capacity, with a dephosphorization rate of 82.32% and a denitrification rate of 73.73%. The ADPB in the A$^2$O aerobic biological pool of the sewage treatment plant at Hezuo Town demonstrated a rich diversity and a strong DPR capacity.

**Keywords:** aerobic denitrifying phosphate accumulating bacteria (ADPB); diversity; denitrifying phosphorus removal

---

## 1. Introduction

Nutrients such as nitrogen and phosphorus in excess have been recognized as the primary cause of eutrophication. Therefore, the key to solving the issue of eutrophication is in the removal of nitrogen and phosphorus from wastewater. The national and local departments have introduced stringent discharge standards for nutrients such as nitrogen and phosphorus. The conventional processes for combined nitrogen and phosphorus removal face several problems, such as insufficient carbon source, the competition of flora, difficulty in controlling sludge age, and reflux of mixed liquid [1,2]. Therefore, it makes great sense to seek an approach to denitrify and remove phosphorus simultaneously. Denitrifying phosphate-removal bacteria (DPB) may utilize nitrate as electron acceptors to complete the two processes of denitrification and uptake excessive phosphorus in concert. The discovery of DPB indicated the direction for the development of denitrifying phosphorus removal (DPR) technique, so that it may overcome the drawbacks of the conventional processes, including insufficient carbon source, the competition of flora, hard to control mud age, and reflux of mixed liquid [3]. Kuba et al. [4] reported the cultivation of a type of facultative anaerobe in the alternating anaerobic-anoxic sequencing batch reactor (SBR). This type of anaerobe, the DPB, could use both $O_2$ and $NO_3^-$ as electron acceptors to remove phosphorus and simultaneously carry out denitrification. In their study of DPR using the anaerobic-anoxic SBR ($A^2$SBR), Ng et al. [5] revealed the existence of DPB in the anaerobic/anoxic/anaerobic biological phosphorus removal system. From their experiment on optimization of anoxic phosphorus removal in the $A^2$SBR, Merzouki et al. [6] demonstrated that the anoxic phosphorus removal depended primarily on the amount of nitrate added and the age of the sludge. Luo et al. [7] used the anaerobic anoxic nitrifying-anaerobic sequencing batch reactor (A2N-ASBR) to culture activated sludge and found that the *Aeromonas*, *Enterobacteriaceae*, *Pseudomonas*, and *Moraxella* played a significant role in DPR. The biological phosphorus release and uptake experiment conducted by Kerrn-Jespersen et al. [8] in an alternating anaerobic-anoxic environment provided by a fixed biofilm reactor, and showed that this process may utilize nitrate as an electron acceptor to achieve the purpose of biological phosphorus removal. Later, Zhang et al. [9] compared the inverted anaerobic/anoxic/oxic ($A^2$O) process with the conventional $A^2$O process and found that, after the addition of an appropriate amount of nitrate in the anoxic stage, the DPB of the conventional $A^2$O process began the uptake of anoxic phosphorus.

During their study of microorganisms for DPR in sewage treatment systems, Ma et al. [10] isolated one strain of DPB, *Bacillus cereus*, from the laboratory A/O/A-SBR. Likewise, Jia et al. [11] isolated four strains of DPB, belonging to *Aeromonas ichthiosmia*, *Pseudomonas stutzeri*, *Citrobacter freundii*, and *Neisseria mucosa*, respectively, from the activated sludge of laboratory $A^2$SBR. Li et al. [12] isolated four strains of DPB, all belonging to *Aeromonas*, from the activated sludge of laboratory SBR. Xiao et al. [13] isolated one strain of DPB, an *Acinetobacter* sp. from the activated sludge of laboratory SBR. Zhu et al. [14] isolated two strains of DPB, belonging to the *Pseudomonas* sp., from the mature activated sludge in a sewage treatment plant. Cai et al. [15] isolated a DPB strain belonging to the *Pseudomonas* sp. using the sludge from the anoxic biochemical pool of a sewage treatment plant as the strain screening subject. Nie et al. [16] isolated two aerobic DPB (ADPB) belonging to the *Achromobacter* sp. and the *Brevundimonas* sp., respectively, from a pig farm sewage treatment pool.

DPR technique focuses on the alternating anaerobic/anoxic environment or on the anoxic stage, while the study on DPR in continuous aerobic conditions is quite limited [17]. The DPR microorganisms from sewage treatment systems were primarily isolated from the laboratory SBR by alternating anaerobic/anoxic domesticating treatment and from the activated sludge in the anoxic stage of wastewater treatment plants. Notwithstanding, the ADPB composition under continuous aerobic conditions has been rarely reported. Therefore, in this study, the activated sludge from the aerated biochemical pool of $A^2$O system was collected, cultured by the pure culture method, enriched in denitrification media, isolated and purified on BTB plates, and screened for ADPB through phosphorus uptake and nitrate reduction assays. Further, through the 16S rRNA gene sequencing, the diversity of

ADPB was analyzed and the DPR capacity was determined in an effort to provide effective sources of strains for the industrial application of DPR.

We propose that a strain of ADPB could be isolated in this study and screened from the activated sludge of the A$^2$O aerated biochemical pool of sewage treatment plants; these ADPB exhibit robust, simultaneous denitrifying and phosphorus accumulating capacities.

## 2. Materials and Methods

### 2.1. Source of the Samples

The activated sludge used for the isolation and purification of ADPB was collected from the A$^2$O aerobic biochemical pool of the sewage treatment plant, Hezuo Town, Chengdu City. After natural agglomeration and concentration, the supernatant was decanted, and the remaining sludge mixture was placed in a sterile polyethylene bag. This was then placed in a foam box containing ice cubes and transported back to the laboratory at the earliest.

### 2.2. Media

Denitrification enrichment media: Sodium succinate (Shanghai Aladdin Biochemical Technology Co., Ltd., Shanghai, China), 2.84 g; NaNO$_3$ (Shanghai Aladdin Biochemical Technology Co., Ltd., Shanghai, China), 10 mM; (NH$_4$)$_2$SO$_4$ (Shanghai Aladdin Biochemical Technology Co., Ltd., Shanghai, China), 0.27 g; KH$_2$PO$_4$ (Shanghai Aladdin Biochemical Technology Co., Ltd., Shanghai, China), 1.36 g; yeast extract (Guangdong Huan Kai Microbiology Technology Co., Ltd., Guangzhou, Guangdong, China), 1 g; MgSO$_4$·7H$_2$O (Shanghai Aladdin Biochemical Technology Co., Ltd., Shanghai, China), 0.19 g; trace element solution (Guangdong Huan Kai Microbiology Technology Co., Ltd., Guangzhou, Guangdong, China), 1 mL, pH 7.2, and distilled water, 1000 mL.

Bromothymol blue (BTB) media [18]: Sodium citrate (Shanghai Aladdin Biochemical Technology Co., Ltd., Shanghai, China), 9.63 g; asparagine (Shanghai Macklin Biochemical Technology Co., Ltd., Shanghai, China), 1 g; KNO$_3$ (Shanghai Aladdin Biochemical Technology Co., Ltd., Shanghai, China), 1 g; KH$_2$PO$_4$ (Shanghai Aladdin Biochemical Technology Co., Ltd., Shanghai, China), 1 g; FeCl$_2$·6H$_2$O (Shanghai Aladdin Biochemical Technology Co., Ltd., Shanghai, China), 0.05 g; CaCl$_2$·2H$_2$O (Shanghai Aladdin Biochemical Technology Co., Ltd., Shanghai, China), 0.2 g; FeCl$_2$·6H$_2$O (Shanghai Aladdin Biochemical Technology Co., Ltd., Shanghai, China), 0.05 g; MgSO$_4$·7H$_2$O (Shanghai Aladdin Biochemical Technology Co., Ltd., Shanghai, China), 1 g; BTB (Guangdong Huan Kai Microbiology Technology Co., Ltd., Guangzhou, Guangdong, China) (1% dissolved in ethanol), 1 mL; agar (Guangdong Huan Kai Microbiology Technology Co., Ltd., Guangzhou, Guangdong, China), 20 g; pH 7.0–7.3; and distilled water (Guangdong Huan Kai Microbiology Technology Co., Ltd., Guangzhou, Guangdong, China), 1000 mL.

Lysogeny broth (LB) media: NaCl (Shanghai Macklin Biochemical Technology Co., Ltd., Shanghai, China), 5 g; peptone (Shanghai Macklin Biochemical Technology Co., Ltd., Shanghai, China), 10 g; agar, 20 g; yeast extract, 3 g; pH 7.2; and distilled water, 1000 mL.

Phosphorus-deficient media: CH$_3$COONa·3H$_2$O (Shanghai Aladdin Biochemical Technology Co., Ltd., Shanghai, China), 3.23 g; Na$_2$HPO$_4$·2H$_2$O (Shanghai Aladdin Biochemical Technology Co., Ltd., Shanghai, China), 23 mg; NH$_4$Cl (Shanghai Aladdin Biochemical Technology Co., Ltd., Shanghai, China), 152.8 mg; MgSO$_4$·7H$_2$O, 81.12 mg; K$_2$SO$_4$, 17.83 mg; CaCl$_2$·2H$_2$O, 11 mg; HEPES(Shanghai Aladdin Biochemical Technology Co., Ltd., Shanghai, China), 7 g; trace element solution, 2 mL; pH 7; and distilled water, 1000 mL.

Phosphorus-rich media: CH$_3$COONa·3H$_2$O, 3.23g; KH$_2$PO$_4$·2H$_2$O, 25 mg; NH$_4$Cl, 305.52 mg; MgSO$_4$·7H$_2$O, 91.26 mg; CaCl$_2$·2H$_2$O, 25.68 mg; HEPES, 8.5 g; trace element solution, 2 mL; pH 7; and distilled water, 1000 mL.

Trace element solution: EDTA (Guangdong Huan Kai Microbiology Technology Co., Ltd., Guangzhou, Guangdong, China), 50 g; MnCl$_2$·4H$_2$O (Shanghai Aladdin Biochemical Technology Co.,

Ltd., Shanghai, China), 5 g; $CuSO_4 \cdot 5H_2O$ (Shanghai Aladdin Biochemical Technology Co., Ltd., Shanghai, China), 1.6 g; $FeSO_4 \cdot 7H_2O$ (Shanghai Aladdin Biochemical Technology Co., Ltd., Shanghai, China), 5 g; $CoCl_2 \cdot 6H_2O$ (Shanghai Aladdin Biochemical Technology Co., Ltd., Shanghai, China), 50 mg; KI (Shanghai Aladdin Biochemical Technology Co., Ltd., Shanghai, China), 10 mg; $(NH_4)_6Mo_7O_{24} \cdot 4H_2O$ (Shanghai Aladdin Biochemical Technology Co., Ltd., Shanghai, China), 1.1 g; $H_3BO_3$ (Shanghai Aladdin Biochemical Technology Co., Ltd., Shanghai, China), 50 mg; and distilled water, 1000 mL.

Media for nitrate reduction experiment [19]: Beef extract (Shanghai Macklin Biochemical Technology Co., Ltd., Shanghai, China), 3.0 g; peptone, 5.0 g; $KNO_3$, 1.0 g; pH 7.0–7.2; and distilled water, 1000 mL.

Nitrogen-phosphorus-rich media: $CH_3COONa \cdot 3H_2O$, 3.23 g; $KH_2PO_4 \cdot 2H_2O$, 50 mg; $NH_4Cl$, 250 mg; $MgSO_4 \cdot 7H_2O$, 91.26 mg; $CaCl_2 \cdot 2H_2O$, 19.39 mg; $KNO_3$, 300 mg; trace elements solution, 2 mL; pH 7.0–7.2; and distilled water, 1000 mL.

## 2.3. Enrichment and Isolation of ADPB Strains

The fresh activated sludge mixture obtained from the aerobic biochemical pool of the sewage treatment plant of Hezuo Town, Chengdu City, was mixed well on a magnetic stirrer. The muddy water mixture was allowed to stand still for 2 h, and 10 mL of the supernatant was transferred by a pipette to a 250 mL Erlenmeyer flask containing 100 mL denitrification enrichment media, sealed with nine layers of gauze, and then enriched in a thermostatic incubator at 28 °C and 160 rpm. After 24 h, 10 mL of the enrichment culture was transferred by a sterile pipette to another 250 mL Erlenmeyer flask containing 90 mL of sterile distilled water and shaken for 30 min to obtain $10^{-1}$ bacterial suspension. Further, 1 mL of $10^{-1}$ bacterial suspension was transferred to 9 mL of sterile water to obtain the $10^{-2}$ bacterial suspension and sequentially diluted to obtain the gradients $10^{-3}$, $10^{-4}$, etc., of the bacterial suspension. Three appropriate gradients ($10^{-4}$, $10^{-5}$, and $10^{-6}$) were selected for cultivation. Then, the cells were plated on BTB plates, and further isolated and purified by dilution plating and streaking methods.

## 2.4. ADPB Strain Screening

The obtained isolated blue colonies were assessed for phosphorus uptake, nitrate reduction, and subject to metachromatic granules staining and poly-β-hydroxybutyrate (PHB) granules staining to screen for ADPB strains with good DPR performance. For the phosphorus uptake experiment, the strain was first inoculated into the phosphorus-deficient media, cultured overnight at 28 °C and 160 rpm, followed by centrifugation at $4000 \times g$ rpm for 15 min to collect the bacterial cells, which were then washed 2–3 times with sterile saline solution. Then, the cells were used to prepare 5 mL bacterial suspension with optical density ($OD_{600}$) of approximately 0.5, inoculated into 50 mL sterilized fresh phosphorus-rich media at 28 °C and 160 rpm for 48 h. After centrifugation at $4000 \times g$ rpm for 15 min, the supernatant was spectrophotometrically analyzed using ammonium molybdate to determine the rate of phosphorus uptake [20]. LB slants of the obtained ADPB strains were prepared and stored in a 4 °C refrigerator.

## 2.5. Genomic DNA Extraction and PCR Amplification of 16S rDNA

The genomic DNA of the strain was extracted by the kit (Tiangen Biotechnology (Beijing) Co., Ltd., Beijing, China), and the universal primers 27F (5′-AGAGTTTGATCCTGGCTCAG-3′) and 1492R (5′-GGTTACCTTGTTACGACTT-3′) were used for the PCR amplification of 16S rRNA [21]. The PCR reaction system (20 μL) was: $10 \times$ Ex Taq buffer, 2.0 μL; 5 U Ex Taq, 0.2 μL; 2.5 mM dNTP Mix, 1.6 μL; 27F, 1 μL; 1492R, 1 μL; DNA, 0.5 μL; and ddH2O, 13.7 μL. The PCR was carried out as follows: Initial denaturation at 95 °C for 5 min, 25 cycles of denaturation at 95 °C for 30 s, annealing at 56 °C for 30 s, extension at 72 °C for 90 s, and with a final extension at 72 °C for 10 min. The PCR amplification products were detected by 0.8% agarose gel electrophoresis and then sent for sequencing to Shanghai Majorbio Bio-pharm Technology Co., Ltd. (Shanghai, China).

## 2.6. Phylogenetic Analysis

The obtained raw sequences were processed using the SeqMan II software (version 15.90, DNASTAR, Madison, WI, USA, 2018) to remove vector and contaminating host sequences, and assembled into contigs. The obtained 16S rDNA full sequences were sent for BLAST analysis at EzBioCloud (https://www.ezbiocloud.net) to find the closest species with the highest similarity to the strains. The similar typical, published strain sequences were also retrieved from the database to perform multiple sequence alignment and sequence editing was carried out using Clustal X (version 2.1, EMBL, Heidelberg, Germany, 2012). The Kimura 2-parameter model in MEGA X (version 10.05, Institute for Genomics and Evolutionary Medicine, Temple University, Philadelphia, PA, USA, 2018) was used to calculate the evolutionary distance. Finally, the 16S rDNA phylogenetic tree was constructed using the Neighbor-Joining (NJ) method with a self-expanding value of 1000 [22].

## 2.7. Study on the Characteristics of Denitrification and Dephosphorization

Each isolated ADPB strain was inoculated in an Erlenmeyer flask containing 50 mL of phosphorus-deficiency media and cultured for 24 h in a shaker at 28 °C and 160 rpm. After centrifuging at $4000\times g$ rpm for 15 min, the supernatant was discarded, and the precipitate was washed 2–3 times with sterile physiological saline. Then, 5 mL bacterial suspension at $OD_{600}$ of approximately 0.5 was prepared and inoculated into 50 mL of freshly sterilized phosphorus-rich media. The cells were cultured in a shaker at 28 °C and 160 rpm for 48 h. After centrifugation at $4000\times g$ rpm for 15 min, the supernatant was subjected to ammonium molybdate spectrophotometry to determine the total phosphorus (TP) content [20]. The total nitrogen (TN) was determined by alkaline potassium persulfate digestion, followed by ultraviolet (UV) spectrophotometric method [23]. For each sample, three replicates were set up, and the media with no cells inoculated was set up as control.

## 2.8. Statistical Analysis

Statistical analysis was performed using SPSS statistical software (version 20.0, IBM, Armonk, NY, USA). The one-way analysis of variance (ANOVA) was used to test the difference between the identified strains. Pearson correlation coefficients between the TN and TP removal rate were determined. Significance levels were set at $p = 0.05$ in all statistical analyses.

## 3. Results

### 3.1. Strain Isolation and Screening

After culturing the collected activated sludge in the denitrification enrichment media, the cell suspension was applied to BTB solid plates through the dilution plating method. After checking the colony properties, single colonies were picked and streaked on plates for isolation and purification. The process resulted in 67 isolated strains. Thirty-three strains were selected from the phosphorus uptake experiment. Further, 25 strains were obtained with DPR capacity after nitrate reduction test, metachromatic granules staining, and PHB staining.

### 3.2. Identification and Analysis of Diversity

The 16S rDNA BLAST results of the screened, cultivatable ADPB are shown in Table 1. The strains with the 16S rRNA gene sequence having greater than 97% similarity were identified as the same species [24]. According to their determined 16S rRNA sequences, the 25 strains were classified into 14 different species. The strain SW18NP2 shared a 16S rRNA gene sequence similarity of 96.50% with the *Acinetobacter johnsonii* strain CIP (Collection de L'Institut Pasteur Of Institut Pasteur) 64.6 (APON01000005), possibly a new species of the *Acinetobacter* genus. The strain SW18NP24 shared a 16S rRNA similarity of 97.00% with the *Pseudomonas simiae* strain OLi (AJ936933), and a potentially new species of the *Pseudomonas* genus.

**Table 1.** The online BLAST results of 16S rRNA gene sequences for the 25 aerobic denitrifying phosphate accumulating bacteria (ADPB) strains in the activated sludge of the A²O aerobic biological pool.

| Bacteria Strain | Nearest Phylogenetic Neighbor (Accession Number) [a] | Gene Identity (%) [b] | Taxonomical Assignment | Accession Number [c] |
|---|---|---|---|---|
| SW18NP1 | *Acinetobacter* sp. strain WCHA55 (NHRN01000069) | 99.06% | *Acinetobacter* sp. | MK861091 |
| SW18NP2 | *Acinetobacter johnsonii* strain CIP 64.6 (APON01000005) | 96.50% | *Acinetobacter johnsonii* | MK861092 |
| SW18NP3 | *Acinetobacter tandoii* strain DSM 14,970 (KE007359) | 98.91% | *Acinetobacter tandoii* | MK861093 |
| SW18NP4 | *Acinetobacter* sp. strain ANC 4173 (SJOF01000009) | 98.77% | *Acinetobacter* sp. | MK861094 |
| SW18NP5 | *Acinetobacter tandoii* strain DSM 14,970 (KE007359) | 99.14% | *Acinetobacter tandoii* | MK861095 |
| SW18NP6 | *Delftia tsuruhatensis* strain NBRC 16,741 (BCTO01000107) | 99.78% | *Delftia tsuruhatensis* | MK861096 |
| SW18NP7 | *Pseudomonas* sp. strain KT2440 (AE015451) | 99.78% | *Pseudomonas* sp. | MK861097 |
| SW18NP8 | *Acinetobacter tandoii* strain DSM 14,970 (KE007359) | 99.28% | *Acinetobacter tandoii* | MK861098 |
| SW18NP9 | *Acinetobacter tandoii* strain DSM 14,970 (KE007359) | 99.20% | *Acinetobacter tandoii* | MK861099 |
| SW18NP10 | *Acinetobacter johnsonii* strain CIP 64.6 (APON01000005) | 99.00% | *Acinetobacter johnsonii* | MK861100 |
| SW18NP11 | *Aeromonas media* strain CECT 4232 (CDBZ01000012) | 99.78% | *Aeromonas media* | MK861101 |
| SW18NP12 | *Acinetobacter oryzae* strain B23 (GU954428) | 99.93% | *Acinetobacter oryzae* | MK861102 |
| SW18NP13 | *Acinetobacter johnsonii* strain CIP 64.6 (APON01000005) | 98.92% | *Acinetobacter johnsonii* | MK861103 |
| SW18NP14 | *Acinetobacter* sp. strain WCHA55 (NHRN01000069) | 98.93% | *Acinetobacter* sp. | MK861104 |
| SW18NP15 | *Citrobacter portucalensis* strain A60 (MVFY01000035) | 99.78% | *Citrobacter portucalensis* | MK861105 |
| SW18NP16 | *Delftia lacustris* strain LMG 24,775 (jgi.1102360) | 99.78% | *Delftia lacustris* | MK861106 |
| SW18NP17 | *Delftia tsuruhatensis* strain NBRC 16,741 (BCTO01000107) | 99.78% | *Delftia tsuruhatensis* | MK861107 |
| SW18NP18 | *Acinetobacter johnsonii* strain CIP 64.6 (APON01000005) | 98.93% | *Acinetobacter johnsonii* | MK861108 |
| SW18NP19 | *Pseudomonas* sp. strain R17(2017) (NEIG01000032) | 99.78% | *Pseudomonas* sp. | MK861109 |
| SW18NP20 | *Acinetobacter johnsonii* strain CIP 64.6 (APON01000005) | 98.99% | *Acinetobacter johnsonii* | MK861110 |
| SW18NP21 | *Acinetobacter johnsonii* strain CIP 64.6 (APON01000005) | 99.00% | *Acinetobacter johnsonii* | MK861111 |
| SW18NP22 | *Acinetobacter bouvetii* strain DSM 14,964 (APQD01000004) | 98.92% | *Acinetobacter bouvetii* | MK861112 |
| SW18NP23 | *Delftia acidovorans* strain 2167 (JOUB01000005) | 99.78% | *Delftia acidovorans* | MK861113 |
| SW18NP24 | *Pseudomonas simiae* strain OLi (AJ936933) | 97.00% | *Pseudomonas simiae* | MK861114 |
| SW18NP25 | *Pseudomonas* sp. strain R17(2017) (NEIG01000032) | 99.78% | *Pseudomonas* sp. | MK861115 |

Note: [a] The sequence with highest percentage of identity observed in EzBioCloud; [b] The percentage of identity with EzBioCloud analysis; [c] The accession number of ADPB in national center for biotechnology information (NCBI).

According to the 16S rRNA gene sequence analysis (Figure 1), all the 25 strains of ADPB belonged to the phylum *Proteobacteria*, involving two classes (*Gammaproteobacteria* and *Betaproteobacteria*), four orders (*Aeromonadales, Enterobacterales, Pseudomonadales,* and *Burkholderiales*), five genera (*Aeromonas, Citrobacter, Pseudomonas, Acinetobacter,* and *Delftia*), and 14 species. The genus *Acinetobacter* had the highest number of ADPB, with 15 strains, followed by the genus *Pseudomonas* and *Delftia*, each with four strains, while the genus *Delftia* and *Citrobacter* had only one strain each of ADPB.

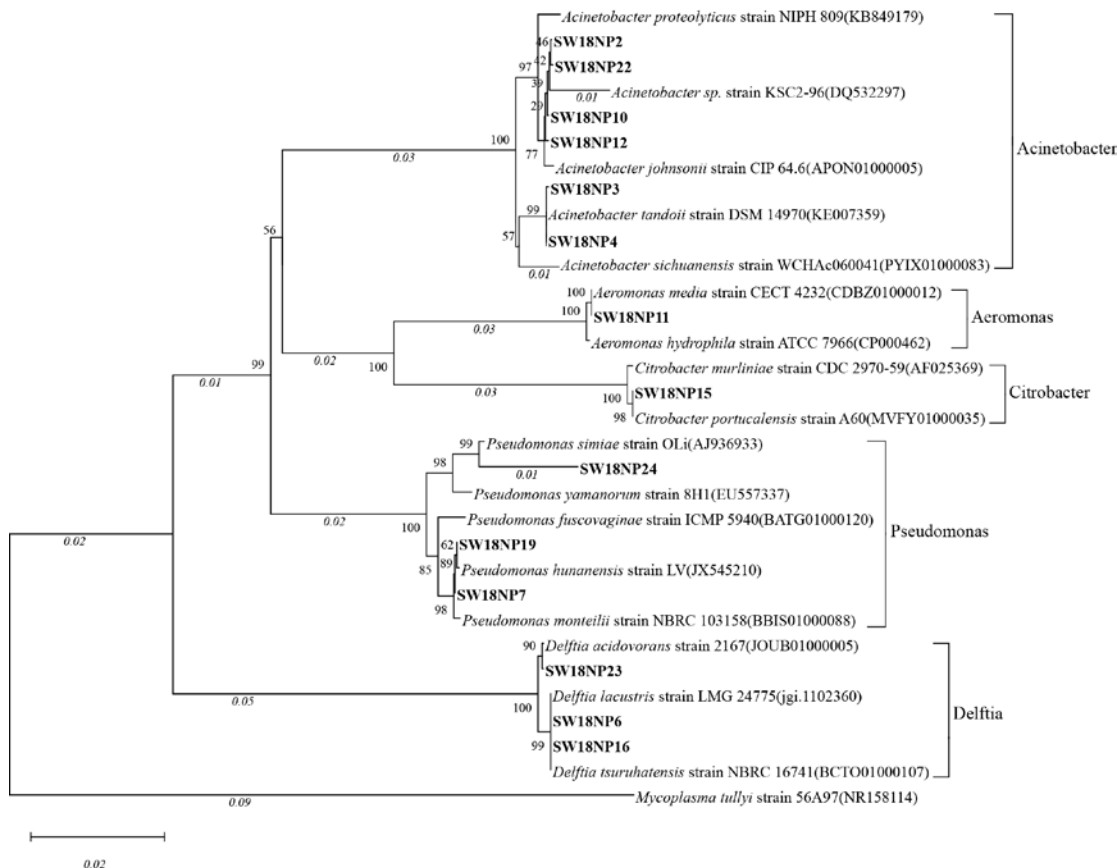

**Figure 1.** A neighbor-joining tree showing the phylogenetic relationships among 16S rDNA sequences of isolated ADPB strains and their closely related sequences from EzBioCloud. The numbers at the nodes indicate the bootstrap values based on the neighbor-joining analyses of 1000 resampled data sets. The scale bar indicates evolutionary distance.

### 3.3. Denitrification and Dephosphorization Analyses

After culturing each of the 25 strains in phosphorus-deficient media to release phosphorus, they were inoculated into the nitrogen-phosphorus-rich media and cultured for 48 h. The resulting pH values, $OD_{600}$, TN, and TP contents of the culture are shown in Table 2.

As shown in Table 2, the TN and TP contents in the supernatant of ADPB culture media were both significantly reduced compared to that in the control ($p < 0.05$). The DPR capacities of most ADPB strains varied significantly among themselves ($p < 0.05$). The TN content in the culture media was 16.06~40.16 mg/L, the denitrification rate was 35.07~74.03%. The TP content was 1.20~4.40 mg/L, and the dephosphorization rate was 35.20~82.32%. Twenty-three strains exhibited above 50% TP removal rate, and 19 strains exhibited above 50% TN removal rate. Among them, the strain SW18NP22 exhibited the highest dephosphorization rate of 82.32%, and the strain SW18NP16 exhibited the highest denitrification rate of 74.03% at 48 h. Considering both denitrification and dephosphorization capacities, the strain SW18NP2 was adjudged as the most efficient, with dephosphorization and denitrification rates of 82.32% and 73.73%, respectively.

**Table 2.** The amounts of P-uptake, amounts of TN-liberated, OD, pH, and time of growth.

| Bacteria Strain | Time of Growth (h) [a] | OD$_{600}$ [b] | pH [c] | TN (mg/L) [d] | TP (mg/L) [e] |
|---|---|---|---|---|---|
| CK | | | 7.0 ± 0.3 [c] | 61.85 ± 0.43 [a] | 6.79 ± 0.13 [a] |
| SW18NP1 | 48 | 0.933 ± 0.012 [b] | 9.5 ± 0.2 [a] | 21.56 ± 0.15 [f] | 2.49 ± 0.09 [de] |
| SW18NP2 | 48 | 0.770 ± 0.009 [b] | 9.5 ± 0.2 [a] | 21.79 ± 0.20 [f] | 1.80 ± 0.11 [f] |
| SW18NP3 | 48 | 0.976 ± 0.008 [b] | 9.5 ± 0.3 [a] | 23.77 ± 0.18 [f] | 1.79 ± 0.09 [f] |
| SW18NP4 | 48 | 0.993 ± 0.011 [ab] | 9.5 ± 0.1 [a] | 24.14 ± 0.11 [f] | 1.48 ± 0.08 [fg] |
| SW18NP5 | 48 | 1.007 ± 0.012 [ab] | 9.4 ± 0.2 [a] | 29.69 ± 0.13 [e] | 2.20 ± 0.09 [e] |
| SW18NP6 | 48 | 0.604 ± 0.009 [b] | 9.4 ± 0.2 [a] | 25.98 ± 0.19 [e] | 1.63 ± 0.12 [f] |
| SW18NP7 | 48 | 1.329 ± 0.008 [a] | 9.5 ± 0.2 [a] | 35.09 ± 0.12 [cd] | 2.86 ± 0.08 [d] |
| SW18NP8 | 48 | 0.978 ± 0.011 [b] | 9.5 ± 0.1 [a] | 21.37 ± 0.20 [f] | 1.83 ± 0.07 [f] |
| SW18NP9 | 48 | 0.965 ± 0.010 [b] | 9.5 ± 0.2 [a] | 20.90 ± 0.17 [f] | 1.74 ± 0.09 [f] |
| SW18NP10 | 48 | 0.944 ± 0.013 [b] | 9.5 ± 0.2 [a] | 29.12 ± 0.09 [e] | 1.88 ± 0.10 [f] |
| SW18NP11 | 48 | 0.532 ± 0.007 [b] | 9.4 ± 0.3 [a] | 33.58 ± 0.21 [d] | 2.97 ± 0.09 [cd] |
| SW18NP12 | 48 | 0.859 ± 0.009 [b] | 9.5 ± 0.2 [a] | 40.16 ± 0.11 [b] | 4.40 ± 0.11 [b] |
| SW18NP13 | 48 | 1.020 ± 0.008 [a] | 9.5 ± 0.3 [a] | 20.81 ± 0.10 [f] | 2.08 ± 0.06 [e] |
| SW18NP14 | 48 | 0.860 ± 0.011 [b] | 9.5 ± 0.3 [a] | 31.42 ± 0.18 [d] | 3.21 ± 0.13 [c] |
| SW18NP15 | 48 | 0.924 ± 0.009 [b] | 9.0 ± 0.2 [ab] | 25.08 ± 0.14 [ef] | 1.24 ± 0.08 [g] |
| SW18NP16 | 48 | 0.571 ± 0.011 [b] | 9.4 ± 0.3 [a] | 16.06 ± 0.14 [g] | 3.69 ± 0.12 [c] |
| SW18NP17 | 48 | 0.620 ± 0.008 [b] | 9.4 ± 0.2 [a] | 24.24 ± 0.17 [f] | 3.36 ± 0.09 [c] |
| SW18NP18 | 48 | 0.949 ± 0.008 [b] | 9.5 ± 0.2 [a] | 24.80 ± 0.18 [f] | 2.57 ± 0.07 [d] |
| SW18NP19 | 48 | 0.555 ± 0.009 [b] | 9.4 ± 0.3 [a] | 39.32 ± 0.11 [c] | 3.23 ± 0.08 [c] |
| SW18NP20 | 48 | 0.972 ± 0.008 [b] | 9.4 ± 0.1 [a] | 26.26 ± 0.14 [e] | 1.63 ± 0.11 [f] |
| SW18NP21 | 48 | 0.975 ± 0.011 [b] | 9.6 ± 0.3 [a] | 21.23 ± 0.19 [f] | 1.77 ± 0.09 [f] |
| SW18NP22 | 48 | 0.890 ± 0.012 [b] | 9.5 ± 0.3 [a] | 16.25 ± 0.18 [g] | 1.20 ± 0.12 [g] |
| SW18NP23 | 48 | 0.633 ± 0.009 [b] | 9.5 ± 0.2 [a] | 23.77 ± 0.16 [f] | 2.97 ± 0.08 [cd] |
| SW18NP24 | 48 | 0.499 ±0.007 [bc] | 9.4 ± 0.2 [a] | 32.55 ± 0.11 [d] | 2.94 ± 0.07 [cd] |
| SW18NP25 | 48 | 0.569 ± 0.007 [b] | 9.5 ± 0.1 [a] | 34.29 ± 0.13 [d] | 3.09 ± 0.01 [c] |

Note: Each value is represented as the mean ± S.E; control (CK): No inoculation. [a]. time of growth (h) in the nitrogen-phosphorus-rich media; [b]. OD$_{600}$ values in nitrogen-phosphorus-rich media at 48 h; [c]. pH of the nitrogen-phosphorus-rich media at 48 h; [d]. TN content in the supernatant of nitrogen-phosphorus-rich media at 48 h; [e]. TP content in the supernatant of nitrogen-phosphorus-rich media at 48 h. In the same column, data with different letters such as [a], [b] and [c] indicate significant differences, while data with one or more same letters indicate insignificant differences at the 0.05 level. For example data with letters [ab] were insignificantly different from data with letters [ab], [a], [b] and [bc]. Data with letters [bc] were insignificantly different from data with letters [bc], [b], [c] and [cd]. Data with letters [de] were insignificantly different from data with letters [de], [d], [e] and [ef]. The rest can be done in the same manner.

The pH of the ADPB culture media were significantly higher than that of the control ($p < 0.05$) (Table 2), and no significant variation was observed in the pH values of the culture media with various ADPB strains ($p > 0.05$). With an initial pH of approximately 7.0, the culture pH reached 9.0~9.6 after 48 h of cultivation, indicating that ADPB strains produced enzymes to reduce nitrate and generate alkaline substances during their growth.

After being cultured for 48 h, the ADPB strains SW18NP7 and SW18NP13 showed high OD$_{600}$ values (Table 2), significantly different from that of the other strains ($p < 0.05$), except SW18NP4 and SW18NP5. Further, other than the strains SW18NP7 and SW18NP13, the OD$_{600}$ values of the other strains did not vary significantly from each other ($p > 0.05$). These results indicate that the growth characteristics of various ADPB strains isolated from the activated sludge from the A$^2$O aerobic biological pool were not significantly different.

Pearson's correlation analysis revealed that the denitrification and dephosphorization rates of ADPB were significantly correlated (r = 0.648, $p < 0.05$). However, the rates of denitrification and dephosphorization capacity were not directly related. For example, although the strain SW18NP16 showed a superior denitrification rate, the dephosphorization capacity was normal.

## 4. Discussion

According to the published reports, *Pseudomonas*, *Aeromonas*, *Acinetobacter*, *Bacillus*, *Citrobacter*, *Neisseria*, *Enterobacter*, *Moraxella*, *Achromobacter*, and *Brevundimonas* are common ADPB isolated from the

environment [7,12,14,16]. In the microorganisms isolated from the activated sludge of the A$^2$N/ASBR bis sludge reactor by Luo et al. [7], four species, *Aeromonas*, *Enterobacter*, *Aeromonas*, and *Moraxella* accounted for 66.6% of the total number of bacteria, and their primary role was DPR. Among them, the *Aeromonas* was the most abundant, accounting for 22.9% of all bacteria, and therefore, was the dominant genus. Zhou et al. [25] studied the sludge from the A$^2$O anoxic biological pool using a phosphorus release/accumulation device and pure culture method to determine the ADPB species in the pool. They found that the strains from *Enterobacteriaceae*, *Aeromonas*, and *Pseudomonas* had aerobic denitrification capability. The ADPB isolated in this study were different from those isolated from the activated sludge of the A$^2$N/ASBR bis sludge reactor by Luo et al. [7] and from the sludge of the A$^2$O anoxic biological pool by Zhou et al. [25], presumably due to different culture environments of these studies. In this study, the ADPB were isolated from a continuous aerobic environment, whereas those isolated by Luo et al. and Zhou et al. were from alternating anaerobic-anoxic-aerobic and anoxic environments, respectively. One of the important factors affecting the type and distribution of ADPB in wastewater treatment systems was the dissolved oxygen (DO). Lotter [26] isolated and purified ADPB from the activated sludge in the aerobic stage of the Bardenpho process of wastewater treatment, and found that 56–66% of ADPB belonged to *Acinetobacter*, and the others were *Pseudomonas* and *Aeromonas*. However, these ADPB only demonstrated their DPR capacities in anoxic conditions. In this study, the 25 ADPB isolated from the activated sludge of the A$^2$O aerobic biological pool belonged to five genera and involved more diverse species. The most abundant genus was *Acinetobacter*, and the genus *Delftia* isolated in this study has never been reported as ADPB before.

Compared with the ADPB isolated by Xie et al. [27] from wetland sediments, the ADPB isolated from this study demonstrated a high DPR activity and a good diversity. Compared with the ADPB isolated by Luo et al. [7] from the activated sludge of the A$^2$N/ASBR bis sludge reactor, those isolated from this study exhibited a high diversity and a high dephosphorization activity.

Liu et al. [28] showed that the ADPB strains isolated by them did not favor an acidic environment, rather preferred an environment with pH ~8. Wang et al. [29] found that when pH was greater than 8, the phosphorus concentration dropped sharply due to chemical precipitation, making it difficult to determine whether the phosphorus was removed by biological activity or by chemical precipitation. Wang et al. [30] used an SBR to study the effect of feed water pH on DPR, with nitrite as an electron acceptor. They showed that pH too high (>8) or too low (<6) were not conducive to the phosphorus absorption of DPB. In this study, all the ADPB strains isolated from the activated sludge of the A$^2$O aerobic biological pool could grow in an environment with pH > 9. Most of these strains showed high DPR rate, probably because ADPB strains grew in a high DO environment, which increased the dissolution of phosphorus precipitated in a high pH environment, while the rates of carbon absorption and phosphorus removal were less affected. Moreover, possibly, the alkalinity caused by ADPB denitrification would lead to phosphate precipitation, which contributed to the phosphorus removal rate of ADPB.

Two different hypotheses regarding the DPR capacity of DPB have been proposed [31]. The first one is the two types of bacteria hypothesis, categorizing the polyphosphate-accumulating organisms (PAOs) in the biological phosphorus removal system into two classes. The first class can only use $O_2$ as an electron acceptor, while the second class can use both $O_2$ and $NO_3^-$ as electron acceptors. Therefore, the second class of bacteria may complete denitrification during phosphorus uptake. The other is one type bacteria hypothesis, which states that there is only one class of PAOs in the biological phosphorus removal system. The bacteria in this class may demonstrate denitrification capacity to a certain extent, and depends critically on the reinforcement of the alternating anaerobic/anoxic environment. Once the environment is reinforced, the denitrification enzymes inside the PAOs get induced, imparting the bacteria with denitrifying capacity. In this study, the ADPB were isolated from the aerobic conditions and cultured in a continuous aerobic environment, and demonstrated a significant correlation between the rates of denitrification and dephosphorization. Therefore, the PAOs obtained from the A$^2$O aerobic biological pool belonged to the category of bacteria which can use both $O_2$ and $NO_3^-$ as electron

acceptors. The isolated ADPB completed the phosphorus uptake and denitrification in concert while using $NO_3{}^-$ as an electron acceptor.

In the conventional DPR system, denitrification occurs only under anaerobic conditions, and the involved denitrifying reductases get inhibited in the presence of $O_2$ [32]. It is generally believed that in an aerobic environment, the denitrifiers preferentially use $O_2$ as the electron acceptor for aerobic respiration, thereby preventing $NO_3{}^-$ as the final electron acceptor [33]. Nevertheless, later it was found that the aerobic denitrifiers could use the periplasmic nitrate reductase (Nap) in the cell for denitrification in an aerobic environment. Electrons could be transmitted to $O_2$ as well as to $NO_3{}^-$, of which the latter could serve as the electron acceptor of PHB oxidation. This meant that two mutually independent processes of denitrification and dephosphorization could be executed at the same time [34]. In this study, the ADPB isolated from the $A^2O$ aerobic biological pool also exhibited simultaneous denitrification and excessive phosphorus uptake using $NO_3{}^-$ as the electron acceptor, achieving "dual-use of one carbon". This greatly solved the issues of differences in sludge age and nitrification liquid reflux between the denitrifiers and PAOs in the conventional $A^2O$ DPR system. The cultivation and enrichment of ADPB to make them the dominant bacteria of the $A^2O$ system provides a new strategy for the improvement of the DPR system.

## 5. Conclusions

A total of 25 ADPB strains were isolated from the activated sludge of the $A^2O$ aerobic biological pool of the sewage treatment plant at Hezuo Town, Chengdu City. Based on the 97% sequence similarity, and as per the 16S rRNA gene sequences, the ADPB strains belonged to two classes, four orders, and five genera. The *Acinetobacter* was the dominant genus. The genus *Delftia*, which has been rarely reported, was isolated as an ADPB here. The strains SW18NP2 and SW18NP24 were potentially new species, indicating that the isolation of ADPB from the $A^2O$ aerobic pool expanded the source of ADPB.

In the experiments to assess the DPR capacity, all the 25 ADPB strains could grow well in aerobic alkaline conditions, and the denitrification and dephosphorization rates were significantly correlated. The ADPB strains had strong DPR capacities, although they varied among the isolates. The ADPB from the activated sludge of the $A^2O$ aerobic biological pool are important source to obtain effective strains for the industrial application of DPR technique.

**Author Contributions:** Y.L. conceived and designed the experiments and wrote the paper; S.Z. and J.Z. (Jiejie Zhang) performed the experiments and contributed the reagents and analysis tools; Y.H., R.G. and J.Z. (Jianqiang Zhang) analyzed the data.

**Funding:** This research was funded by the Sichuan science and technology support project (2018GZ0416).

**Acknowledgments:** The authors gratefully acknowledge the technical support of W.X.W. in the sewage treatment plant at Hezuo Town, Chengdu City.

**Conflicts of Interest:** The authors declare no conflict of interest.

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
