# Peer review of "Screening and Diversity Analysis of Aerobic Denitrifying Phosphate Accumulating Bacteria Cultivated from A2O Activated Sludge"

_processes, doi:10.3390/pr7110827_

Round 1

Reviewer 1 Report

This manuscript is a study to locate candidate ADPB strains. From the methodological point of view, I believe that the study is correct, does not contain methodological errors and the results are consistent with the methodology.
Mainly the manuscript contains numerous typographical errors, some of which I indicate:

1) Replace "medium" with "culture media" (plural).

2) Replace mMoL with mM.

3) Superscript formats have been lost.

4) I recommend mL (with uppercase L), in the manuscript there are cases that are written with lowercase and others with uppercase. Unify

5) A large number of genera and species are written without italics. Check and correct.

6) The format of the bibliography should be revised to be more homogeneous, since missing points, commas, italics, spaces, etc.

In addition to this small bugs, I would consider the following improvements:

1) Include in Material and Methods which statistical analyzes were applied to compare the significance of the data.

2) Discussion, I think the discussion is too general and in my opinion it would be appropriate to focus this discussion more on the strains found. For example, comparing with other strains would be a good option for the audience.

Author Response

Dear Editors and Reviewers:

Revised portion are marked using the "Highlight" function in the paper. The main corrections in the paper and the responds to the reviewer’s comments are as flowing:

Responds to the reviewer’s comments:

Comments and Suggestions for Author1

1) Replace "medium" with "culture media" (plural)

Reply: "medium" has been modified into "media"

2) Replace mMoL with mM

Reply: Accepted and it is changed.

3) Superscript formats have been lost

Reply: Superscript formats have been supplemented

4) I recommend mL (with uppercase) in the manuscript there are cases that are written with lowercase and others with uppercase. Unify

Reply: Accepted and it is changed.

5) A large number of genera and species are written without italics. Check and correct.

Reply: Accepted and it is checked and corrected.

6) The format of the bibliography should be revised to be more homogeneous, since missing points, commas, italics, spaces, etc.

 Reply: Accepted and it is revised.

In addition to this small bugs, I would consider the following improvements:
1) Include in Material and Methods which statistical analyzes were applied to compare the significance of the data.

 Reply: "statistical analyses" have been included in Materials and Methods.

2) Discussion, I think the discussion is too general and in my opinion it would be appropriate to focus this discussion more on the strains found. For example, comparing with other strains would be a good option for the audience.

Reply: Accepted and it is changed.

Reviewer 2 Report

The manuscripts provided by Li and co-workers on isolation of 25 ADPB strains from an  activated sludge of the A2O aerobic biological pool of the sewage treatment plant at Hezuo Town, Chengdu City provides very interesting results comparison with published data. English language should be improved, because it is difficult to follow the text. For example, the authors should modified the following sentences:

After the sentence: “According to the available literature [xxx],….” References should be added. In the sentence “As shown in Table 2, the pH” after pH “values” should be added. Instead of bigger in the sentence: “…when pH was bigger than 8, the…” the authors should use “higher” This part is difficult to understand, grammar mistakes… The DPR capacities of most ADPB strains were significant different between them (P < 0.05). The TN content in the culture medium was 16.06~40.16 mg/L, the denitrification rate was 35.07~74.03 %, the TP content was 1.20~4.40 mg/L, and the dephosphorization rate was 35.20~82.32 %. There was a total of 23 strains with TP removal rate of above 50 %, and a total of 19 strains with TN removal rate of above 50 %. A

In my opinion the text should be corrected by an English native speaker or editorial office and then the manuscript could be re-considered for publication. The authors carried out a great job; however, they have not presented it as an easy to follow article.

Author Response

Dear Editors and Reviewers:

Revised portion are marked using the "Highlight" function in the paper. The main corrections in the paper and the responds to the reviewer’s comments are as flowing:

Responds to the reviewer’s comments:

Comments and Suggestions for Author2

The manuscripts provided by Li and co-workers on isolation of 25 ADPB strains from an activated sludge of the A2O aerobic biological pool of the sewage treatment plant at Hezuo Town, Chengdu City provides very interesting results comparison with published data. English language should be improved, because it is difficult to follow the text. For example, the authors should modified the following sentences:

1) After the sentence: “According to the available literature [xxx],….” References should be added.

Reply: Accepted and it is changed.

2) In the sentence “As shown in Table 2, the pH” after pH “values” should be added.

Reply: Accepted and it is changed.

3) Instead of bigger in the sentence: “…when pH was bigger than 8, the…” the authors should use “higher”

Reply: Accepted and it is changed.

4) This part is difficult to understand, grammar mistakes… The DPR capacities of most ADPB strains were significant different between them (P < 0.05). The TN content in the culture medium was 16.06~40.16 mg/L, the denitrification rate was 35.07~74.03 %, the TP content was 1.20~4.40 mg/L, and the dephosphorization rate was 35.20~82.32 %. There was a total of 23 strains with TP removal rate of above 50 %, and a total of 19 strains with TN removal rate of above 50 %.

Reply: Accepted and this part has been revised.

5) In my opinion the text should be corrected by an English native speaker or editorial office and then the manuscript could be re-considered for publication. The authors carried out a great job; however, they have not presented it as an easy to follow article.

Reply: Accepted and the text has been corrected by an English native speaker.

Round 2

Reviewer 2 Report

The corrected manuscript could be accepted for publication.